# Abnormal Development of *Hyalomma Marginatum* Ticks (Acari: Ixodidae) Induced by Plant Cytotoxic Substances

**DOI:** 10.3390/toxins11080445

**Published:** 2019-07-26

**Authors:** Alicja Buczek, Katarzyna Bartosik, Alicja M. Buczek, Weronika Buczek, Dorota Kulina

**Affiliations:** 1Chair and Department of Biology and Parasitology, Medical University of Lublin, 11 Radziwillowska St., 20-080 Lublin, Poland; 2Department of Basic Nursing and Medical Teaching, Medical University of Lublin, Staszica St. 4-6, 20-081 Lublin, Poland

**Keywords:** *Hyalomma marginatum*, abnormal development, morphologic anomalies, plant cytotoxic substance, colchicine, tick control

## Abstract

The increasing application of toxic plant substances to deter and fight ticks proves the need for investigations focused on the elucidation of their impact on the developmental stages and populations of these arthropods. We examined the course of embryogenesis and egg hatch in *Hyalomma marginatum* ticks under the effect of cytotoxic plant substances. The investigations demonstrated that the length of embryonic development of egg batches treated with 20 μL of a 0.1875% colchicine solution did not differ significantly from that in the control group. Colchicine caused the high mortality of eggs (16.3%) and embryos (9.7%), disturbances in larval hatch (8.1%), and lower numbers of normal larval hatches (65.6%). In 0.2% of the larvae, colchicine induced anomalies in the idiosoma (67.6%) and gnathosoma (22.5%) as well as composite anomalies (8.5%). The study demonstrates that cytotoxic compounds with an effect similar to that of colchicine can reduce tick populations and cause teratological changes, which were observed in the specimens found during field studies. Since there are no data on the toxic effects of active plant substances on other organisms and the risk of development of tick resistance, a strategy for the use of such compounds in tick control and the management of plant products should be developed.

## 1. Introduction

Representatives of the genus *Hyalomma*, which are widespread in the area of the subtropical, tropical, and equatorial climate zone [1,2], play an importance role in the epidemiology of tick-borne diseases [3,4,5,6,7,8,9,10].

The great medical and veterinary importance of these ticks and the forecasts of an increase in their spread range [11,12] have been arousing increasing interest in the various aspects of the harmful effects exerted by ticks on the host and the impact of chemical and environmental factors on tick development and population size.

Previous investigations of the biology of the ticks from the genus *Hyalomma* were focused on the feeding patterns [13,14], the life cycle [5,15,16,17], the effect of humidity and temperature on the development of various stages [5], and the course of embryogenesis and larval hatch [18,19]. However, little is known about the effect of chemical factors on ticks in the non-parasitic developmental stages, including the embryonic development and larval hatch. It is advisable that such investigations should be undertaken, since some chemical compounds in the environment that are commonly used for plant protection and control of parasitic arthropods or industrial waste may impede the development of living organisms. The use of plant materials may be an alternative method for reducing tick populations in contrast to the application of commercial acaricides, to which ticks may develop resistance.

Elucidation of the mechanisms of the cytotoxic effect of chemical compounds on ticks in all developmental stages, including embryogenesis, will facilitate the determination of the range of their harmful impact on these arthropods and the identification of the causes of teratological changes observed in specimens collected in nature.

The aim of the study was to examine the effect of microtubule inhibitors on the embryonic development and larval hatch in *H. marginatum* illustrated by the action of colchicine. This is the first study on the effect of plant chemical compounds on embryonic development and larval hatch in this tick species.

## 2. Results

After oviposition, the eggs of *H. marginatum* were spherical. This shape was retained in the first stage of embryonic development, i.e., during cleavage and blastoderm formation, and during gastrulation, which occurred between 1 and ca. 14–16 days after oviposition at a temperature of 25 °C and 75% humidity. During organogenesis, the eggs had an oval shape. The embryonic development and hatches of *H. marginatum* larvae at 25 °C and 75% relative humidity lasted from 28 to 39 days. The duration of larval development, i.e., from 28 to 40 days, was similar in the experiments with the colchicine treatment. As many as 16.3% of the *H. marginatum* eggs died already within the first two days after the administration of colchicine (Table 1). Dead eggs had an altered shape and color. 9.7% of embryos exhibited the inhibition of development in different stages of embryogenesis, and 8.1% of the larvae hatched abnormally. The abnormally hatching larvae were unable to emerge from eggshells or to straighten their legs after emergence, which impaired the motor function. During the hatch, the eggshell started to crack at the posterior part of the idiosoma. Various parts of the larval body remained in the eggshells, mostly all legs or legs on one side of the idiosoma and the gnathosoma. Larvae with disturbed hatching died shortly. Morphological abnormalities were found in 0.2% of the larvae. They hatched from eggs laid by different *H. marginatum* females. Most of the abnormal larvae displayed monstrosities in the idiosoma (67.6%) (Table 2) with a predominance of heteromorphosis of leg segments (deformation of appendages) and oligomely (complete absence of appendages). Heterosymely (fusion of appendages on the same side of the idiosoma) as well as other anomalies of the legs such as an altered shape and size of segments, absence of some segments, atrophy (reduction in the length of appendages), and schistomely (branching appendage) were detected less frequently. The idiosoma had an altered shape in one larva. Morphological anomalies in the gnathosoma constituted 22.5% of all the monstrosities (Table 2). They included changes in the shape and size of chelicerae as well as oligomely of the palp and its abnormal position on the basis of capituli.

Moreover, two concurrent abnormalities referred to as composite anomalies were visible in 8.5% of the larvae (Figure 1 and Figure 2). These were deformations of various gnathosoma or idiosoma parts, sometimes occurring concurrently in both these body parts. They included oligomely, atrophy, heterosymely, deformations of various structures, and asymmetry or changes in the shape of the body.

While there were differences in the number of dead eggs, dead embryos, and abnormally hatched larvae between the colchicine experiments and the control group; the statistical test did not confirm their significance.

In our experiments, there were only 65.6% of morphologically normal larvae. The percentage of normally developed larvae was higher (87.8%) in the control group, but the difference was not statistically significant.

## 3. Discussion

The increase in tick resistance to synthetic chemical compounds and the high costs of tick control necessitates a search for new alternative methods for the reduction of tick populations and the risk of attacks by these arthropods. Therefore, increasing interest in the use of plant substances to deter and fight ticks has been observed in recent years [20,21,22,23,24,25,26,27,28]. The application of plant substances as repellents and/or acaricides for tick control on animals and in habitats is supported by the common opinion that they exhibit lower toxicity to mammals than that of commercial acaricides. They are also characterized by rapid degradation in the environment, which probably has a lower potential of the development of tick resistance, and a more environmentally friendly character in comparison with synthetic chemicals [24]. However, there are still no comprehensive pharmacokinetic studies on the dynamics of absorption of active plant substances as well as their spread to various organs, metabolism, and excretion. The toxic effects of the activity of plant substances and their metabolites on various organisms present in the same ecosystem, including beneficial arthropods (e.g., bees) and mammals, have not been identified either. The suppositions about the absence or low levels of toxicity of natural products require verification in laboratory and field studies. For instance, in contrast to the anecdotal opinion about the low toxicity of leaf aqueous extracts of *Lippia javanica* (family Verbenaceae), i.e., a popular medicinal and garden plant in tropical Africa, the extracts at the dose of 12.5%–37.5% *v*/*v* induced lethargy and eventually the death of 37.5% of examined BALB/c mice. These extracts at higher doses may also be harmful to humans and larger animals [29]. Herbal products containing naturally occurring toxins are implicated in dermatitis as well as many psychological and neurological effects in humans and animals [30].

Our experiments confirm that colchicine, i.e., a cytotoxic alkaloid of plant origin, causes the death of eggs and embryos, disruption of larval hatching, and the development of morphological anomalies in larvae. Similar teratological effects may be caused by other chemical compounds that are present in heavily contaminated environments and are characterized by a similar mechanism of action to that of colchicine on eggs and early embryonic stages in *H. marginatum*. This is implied by the increased frequency of tick developmental disturbances and morphological anomalies in these arthropods observed in regions with a high anthropopressure impact [31,32,33,34,35,36]. Conversely, tick anomalies are rarely observed in unpolluted areas [37,38,39,40].

Through the inhibition of microtubule polymerization caused by binding to tubulin, i.e., one of the main constituents of microtubules, colchicine interrupts microtubule dynamics and disturbs the course of mitosis. Disturbances in embryonic cell divisions in the early embryogenesis stage may lead to the loss and shift of the embryonic material. In this stage, there is considerable translocation of cells connected with the migration of cleavage nuclei towards the egg surface, aggregation of the embryonic material at the egg peripheries, and formation of the blastoderm and the germ band, which undergoes metamerization [19]. As demonstrated by Aeschlimann [41,42], El Kammah et al. [43], Aeschliman and Hess [44], Balashov [45], El Gohary et al. [46], and Shiraishi et al. [47] embryogenesis proceeds similarly in different tick species. Differences between particular species are observed in the length of individual stages of embryogenesis and mechanisms of germ band contraction.

Colchicine-induced disturbances in cell divisions occurring in the early stage of embryonic development result in such monstrosities as oligomely, atrophy, heterosymely, schistomely, and polymely. The pathological consequences of the deleterious activity of this alkaloid also include high proportions of dead eggs and embryos, as well as abnormal hatch of *H. marginatum* larvae.

Disturbances in the course of embryonic development and larval hatch as well as morphological anomalies (atrophy, polymely, schistomely, and heteromorphosis) induced by chemical compounds, e.g., colchicine [33], iodine [48], and lithium [49] compounds, were previously observed in *Argas reflexus*. The experimental use of chemical compounds also caused leg schistomely in *Ornithodoros porcinus* larvae [50] and leg polymely (an additional, fourth pair of legs) in *Dermacentor silvarum* [51]. Different types of local anomalies and disturbances in the embryonic development of the Palearctic *Dermacentor reticulatus* and *Ixodes ricinus* ticks can be induced by synthetic pyrethroids, such as deltamethrin, cypermethrin, alfacypermethrin, and permethrin, which are used for arthropod control [52,53,54]. In addition to the local anomalies described above, chemical compounds can induce general anomalies. Oliver and Delfin [55] obtained two *Dermacentor occidentalis* specimens with gynandromorphism in a generation originating from normal females mating with apholate-treated males. 

The same anomalies were also evoked by high humidity, which is unfavorable for the embryonic stage in this species. It resulted in high embryonic mortality (13.0%) and numerous disturbances in larval hatches (6.9%) [19]. High humidity levels accompanying embryogenesis have been reported to induce teratological changes in species that prefer lower ranges of this parameter, e.g., in the pigeon tick *Argas reflexus* [56]. Disturbances in the embryonic development of arthropods were also evoked by other physical factors, for example temperature [18,57,58,59]. Higher temperature affecting *A. reflexus* embryos was found to result in symely, heterosymely, polymely, and schistomely in harvestmen (Opiliones) [60,61,62] and spiders (Aranea) [63,64,65,66]. A similar effect was obtained after centrifugation of mature spider females and eggs [67,68,69]. Irrespective of the mechanisms of action of various teratogenic factors, disturbances in early embryogenesis may lead to similar morphological anomalies in arthropods. Their frequency clearly varies in the investigated arthropod groups.

Abnormal tick development was observed after repeated infestation of the same hosts. Not only prolonged feeding, reduced body weight, and reduced reproductive potential [70,71,72], but also morphological changes were detected in subsequent developmental stages in ticks [71,73]. Most often, however, the causes of anomalies in ticks in laboratory cultures [74,75,76,77,78,79] and in nature [37,38,39,80,81,82,83,84,85,86,87,88,89] are still undetermined.

Changes in the external structure in ticks may be accompanied by changes in their organs affecting the physiological functions, which, however, have not been studied in this work. In polluted regions of Russia, *Borrelia burgdorferi*-infected specimens of *Ixodes persulcatus* with exoskeleton anomalies displayed 1.3-fold higher locomotor activity than infected ticks without anomalies [90]. This may result in the higher aggression of abnormal females and a greater risk of human infection with pathogens transmitted by the ticks. The presence of tick-borne pathogens (tick-borne encephalitis virus, several species of *Borrelia, Ehrlichia muris*, and *Babesia microti*) in *I. persulcatus* with exoskeletal anomalies was detected in the Kokkola coastal region in Finland (64° N, 23° E) [35]. As shown by Žygutienè et al. [91], the prevalence of tick-borne pathogens (*Borrelia afzelii, Borrelia garinii,* and *Ehrlichia muris*) in anomalous *Ixodes ricinus* ticks collected in the city parks of Vilnius in South-Eastern Lithuania (54°41′ N 25°17′ E) was 1.8 times higher than in normal ones. This value in the case of *Borrelia* spirochetes was 2.8 times higher. Abnormal ticks are characterized by probably having more intense pathogen replication and a higher prevalence of multi-infection with tick-borne pathogens [91,92]. In turn, the encephalitis virus was detected 2.2 times less frequently in ticks with exoskeletal anomalies collected in the Altai Republic in the West Siberia region of Asian Russia (50°55′ N 86°55′ E) than in normal specimens [36].

## 4. Conclusions

The anomalies found in tick specimens collected in natural conditions and in those from the experimental investigations suggest that they may have been caused by chemical compounds. As shown in the study, cytotoxic compounds can decrease the abundance of tick populations through the lethal effect on eggs, inhibition of egg development in various embryogenesis stages, and disturbances in larval hatch. It seems that the use of plant products with cytostatic activity can be an alternative to synthetic chemical compounds for protection of humans and animals against tick bites. However, the application of plant active substances for tick control requires further long-term biological, chemical, and toxicological research.

## 5. Materials and Methods

Adult *H. marginatum* ticks from a Syrian population were reared in laboratory conditions at 25 °C and 75% humidity. Throughout the rearing period, adult ticks and larvae were fed on tick-naive rabbits (*Oryctolagus cuniculus*) [93] in compliance with ethical principles approved by the Local Ethics Committee. In the present study, 12 *H. marginatum* females were randomly selected from 45 engorged specimens, which fed on three hosts with 45 males. To obtain engorged females, 15 hungry females and males were placed on each rabbit. The method of feeding ticks on rabbits was described by Buczek et al. [19,52]. Immediately after drop-off, engorged females were placed individually in chambers. In each experiment, the oviposition process was checked daily until a complete egg batch was laid by the females. One batch comprised from 1580 to 8522 eggs. The egg batches were transferred to the rearing chambers carefully so as not to alter their structure. They were kept at a temperature of 25 °C and 75% humidity, which are optimal for this stage [18], until completion of embryonic development. The required humidity was maintained with the use of a saturated K_2_C_4_H_4_O_6_ solution [94]. On day 2 and 3 after oviposition, each tick egg batch oviposited by nine *H. marginatum* females was carefully placed with a pipette in 20 µL of a 0.1875% aqueous colchicine solution. Previously, a similar concentration of colchicine had been reported to disturb egg and larval hatch in *A. reflexus* [33].

The egg batches were checked every day to observe the course of embryogenesis. After completion of the experiment, the samples were submerged in 70% alcohol and viewed under a stereoscopic microscope to observe the number of dead eggs, dead embryos, abnormally hatching larvae, larvae with morphological abnormalities, and normal larvae. In the control group, three *H. marginatum* egg batches were kept in the rearing chambers at 25 °C and 75% RH and were not treated with colchicine. In the control group, 20 µL of water were applied to each egg batch instead of the colchicine solution.

The morphology of larvae with anomalies and developmental disturbances was examined using light and scanning electron microscopy (SEM). For the SEM investigations, the specimens were cleaned as described by Corwin et al. [95], dehydrated in alcohol, dried in a desiccator, and coated with gold. Classification of the types of anomalies in the *H. marginatum* larvae followed the criteria used by Buczek [19] in studies of tick teratogenesis. Both studies were conducted at the same time.

Statistical analysis. The chi-square test (2 × 2 tables) (*p* ≤ 0.05) was used to assess significant differences between the experimental and control groups.

### Ethical Statement

The laboratory study was performed by Alicja Buczek at the Medical University of Silesia, where she worked in the past (i.e., in 1988–1993). At that time, there were no Ethics Commissions; hence, no written consent for experiments on laboratory animals was mandatory. We do declare, however, that we took care of the welfare of animals participating in the research, which was conducted in conditions ensuring unlimited access to water and food and minimizing animal discomfort. The investigations were conducted as part of scientific activities of the Department and were fully approved by the University authorities.

## Figures and Tables

**Figure 1 toxins-11-00445-f001:**
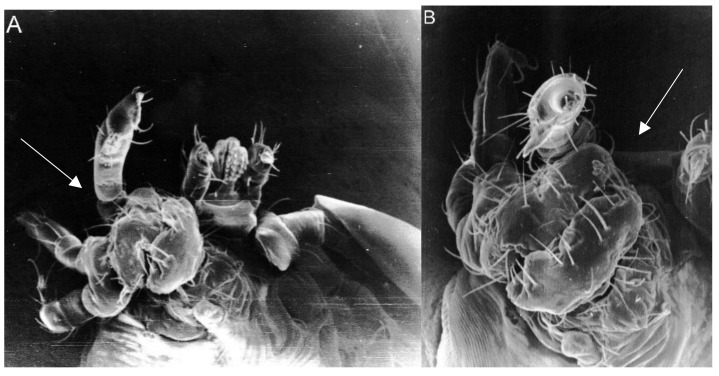
Complex anomaly in a *Hyalomma marginatum* larva. (**A**) Heterosymely of three right legs and heteromorphic leg segments, ventral view (SEM, 350×). (**B**) Fused segments of three legs and an additional appendage on tarsus I, lateral view (SEM, 700×). (**C**) Atrophic left leg III with a deformed coxa only (SEM, 400×). (**D**) Two setae on the left anal valve (SEM, 2500×).

**Figure 2 toxins-11-00445-f002:**
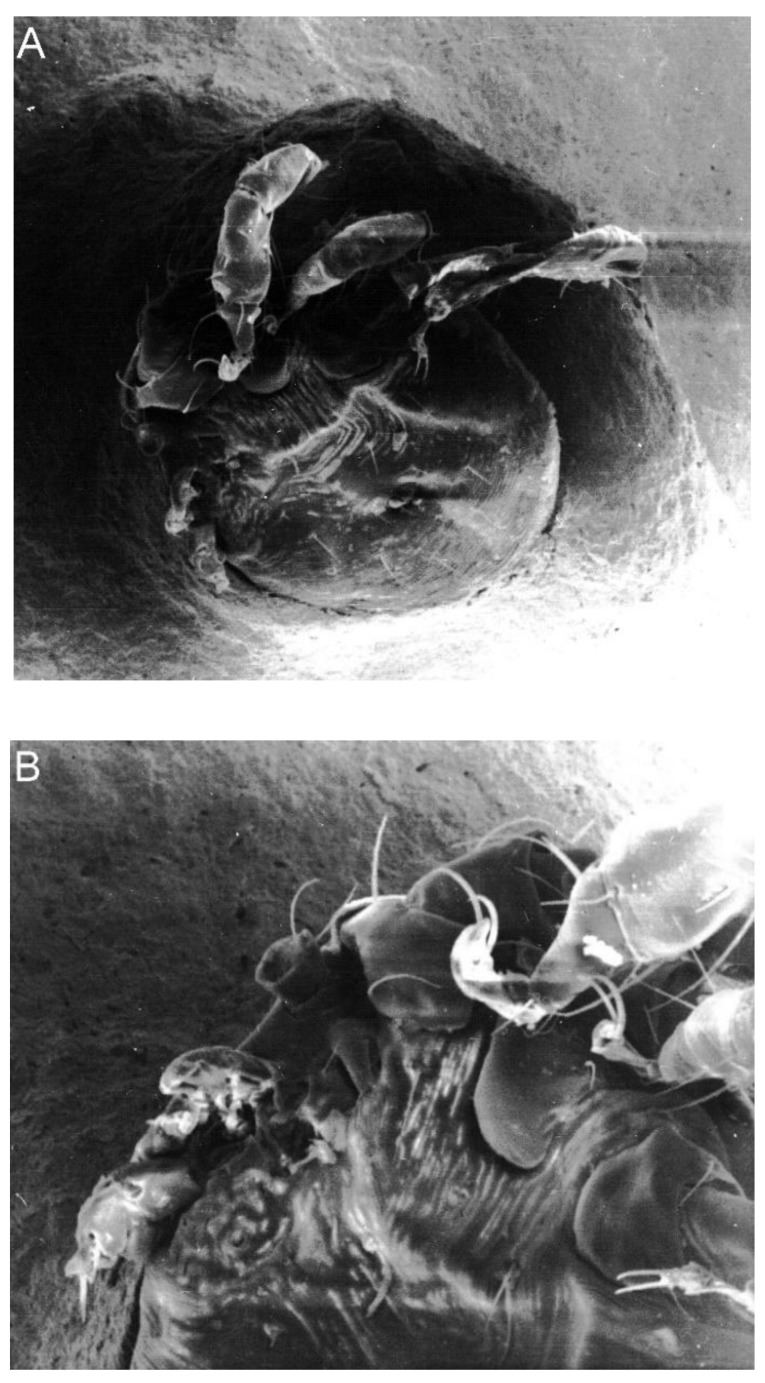
Different types of anomaly in a *Hyalomma marginatum* larva. (**A**) Body asymmetry and oligomely of three right legs (SEM, 250×). (**B**) Anomalies in the gnathosoma: An altered shape of the basis capituli, oligomely of palps, chelicerae, and hypostome (SEM, 400×).

**Table 1 toxins-11-00445-t001:** Embryonic development and egg hatch of *Hyalomma marginatum* under the influence of colchicine at 25 °C and 75% relative humidity (results in percentage) (*n* = 45 779).

Developmental Stages	Examined Group	Control ^a^	*p* Values	Significance
No. of eggs examined	41,512	4267		
Dead eggs	16.3	11.0	0.4142	no
Dead embryos	9.8	0.6	0.5930	no
Abnormally hatched larvae	8.1	0.6	0.6698	no
Larvae with anomalies	0.2	-	0.7150	no
Normal larvae	65.6	87.8	0.7456	no

^a^ Control: eggs kept at 25 °C and 75% RH and not treated with colchicine.

**Table 2 toxins-11-00445-t002:** Monstrosities in *Hyalomma marginatum* larvae induced by colchicine at 25 °C and 75% RH (*n* = 71).

Type of Anomalies	Anomalies	
	Number	%
Gnathosoma	16	22.5
Oligomely of the palp	5	7.0
Atrophy of the palp	1	1.4
Deformation of the palp	3	4.2
Abnormal position of the palp	1	1.4
Elongation of chelicerae	5	7.0
Deformation of chelicerae	1	1.4
Idiosoma	49	67.6
Changed shape of the idiosoma	1	1.4
Oligomely of legs	11	15.5
Atrophy of legs	2	2.8
Heterosymely of legs	3	4.2
Schistomely of legs	1	1.4
Deformation of segments	25	35.2
Atrophy of segments	2	2.8
Other leg anomalies	4	5.6
Composite anomalies	6	8.5
Total	71	100

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
