# Peer review of "Abnormal Development of Hyalomma Marginatum Ticks (Acari: Ixodidae) Induced by Plant Cytotoxic Substances"

_toxins, 2019, doi:10.3390/toxins11080445_

Round 1

Reviewer 1 Report

I have a few comments about the data presentation.

Table 1: please list the P values and the statistical testes used.

Figure 1 and Figure 2 have the same title. Consider changing the title or combining the figures.

Figure 1 and Figure 2: consider using arrows to point/highlight the area of anomaly. 

Author Response

Dear  Reviewer 1,

Thank you very much for your kindness and valuable comments, which we have addressed in the revised version of the manuscript (please see the attachment). 

We wish to explain that we applied colchicine to oviposited H. marginatum eggs. Therefore, we did not assess the course of oviposition or fecundity of females.

The aim of our study was to investigate the impact of colchicine on embryogenesis and hatching of H. marginatum larvae. We have been rearing this species at 25oC and 75% humidity for a long time; hence, we are familiar with the course of embryogenesis in these conditions. In the experimental and control groups used in the present study, the numbers of dead eggs, dead embryos, and abnormally hatched larvae were similar. We have completed the data requested by the Reviewer, i.e. we have specified the number of females examined in the study and the number of eggs per one hatch (oviposited by one female). The number 0.17% (n = 71) is the percentage of abnormal larvae in the experimental group, where n = 41.512 (100%). The discrepancies between the number shown in the text and table 1 are associated with the rounding up of the number. We have unified these values throughout the paper (in red).

In table 1, the percentages are calculated in relation to the number of eggs analysed in the experimental group (n = 41,512) and the controls (n = 4,267). We wish to explain that the statistical analysis presented in Table 1 was carried out during the experiment period in the early 1990s.

To comply with the Reviewer's suggestion, we turned to a professional statistician who showed us a different test, which is more appropriate forthis type of research.We carried out another statistical analysis using the recommended test, and its results have been presened in the revised version.

Table 2 shows the different types of anomalies induced by colchicine. We expressed the share of each type of anomalies among all the observed abnormalities in percentage (n = 71). Since we did not find specimens with anomalies in the control group, there is no column for the control in Table 2.

As suggested by the Reviewer, we expressed the information in lines 156-169 in a concise form.

The selection of the dose of colchicine was prompted by the results of previous similar studies of Argas reflexus. The application of the same doses allowed us to compare the effect of this substance on different species of ticks.

Water was the solvent for colchicine; therefore, only water was applied to the control eggs.

Sincerely Yours

Reviewer 2 Report

This paper investigates developmental defects in a tick species after application with a plant toxin. The investigators carry out a simple exposure experiment on eggs after oviposition and examine the effect on offspring. Eggs exposed to the plant compound have offspring with significantly more defects.

I will not review grammatical or English language style / errors are these are numerous throughout.

Some major issues need to be clarified in the material and methods.

General comments: 

In the introduction I would like further explanation as to the relevance of the investigation and potential control uses of the toxin or why this research may have impact ?

In results i feel you could get a lot more out of the data. Can you comment on the egg laying rate or fecundity of adults in control vs treated ? In table 1 can you put a column with the P values and significance yes/no. Was there a difference in egg numbers laid ? Why are the control numbers so few compared to the treated eggs and could the lack of abnormalities be a function of the far smaller number examined ? How many adults were in the experiment ? What was the mean number of eggs laid ? The percentages are confusing as presumably they are not all a function of eggs examined but of larvae hatched etc ?

In table 2 please put a column for control as well. You state that total is 71. Can you explain how you got the that number from 41512 eggs examined and 0.17% larvae abnormal ?

In the discussion on line 116 can you cite the "opinion" about low toxicity.

Remove lines 156 -169 or condense significantly as this has no immediate relevance to the current study.

In M&M please add more info on ethics committee (line204). Can you comment on the dose of colchicine ? is this field relevant / treatment relevant dose ? How did you come to this conclusion ? What solvent was colchicine dissolved in and was this also used in control group ? 

Were controls treated in a mock manner by moving into a non-toxic solution ? You state that in order to reduce mechanical damage controls had water applied but why were colchicine treated eggs not treated similar ? Please clarify. Move 216 - 219 into the paragraph above.

Author Response

Response to

Dear  Reviewer,

Thank you very much for your kindness and valuable comments, which we have addressed in the revised version of the manuscript. We wish to explain that we applied colchicine to oviposited H. marginatum eggs. Therefore, we did not assess the course of oviposition or fecundity of females.

The aim of our study was to investigate the impact of colchicine on embryogenesis and hatching of H. marginatum larvae. We have been rearing this species at 25oC and 75% humidity for a long time; hence, we are familiar with the course of embryogenesis in these conditions. In the experimental and control groups used in the present study, the numbers of dead eggs, dead embryos, and abnormally hatched larvae were similar. We have completed the data requested by the Reviewer, i.e. we have specified the number of females examined in the study and the number of eggs per one hatch (oviposited by one female). The number 0.17% (n = 71) is the percentage of abnormal larvae in the experimental group, where n = 41.512 (100%). The discrepancies between the number shown in the text and table 1 are associated with the rounding up of the number. We have unified these values throughout the paper (in red).

In table 1, the percentages are calculated in relation to the number of eggs analysed in the experimental group (n = 41,512) and the controls (n = 4,267). We wish to explain that the statistical analysis presented in Table 1 was carried out during the experiment period in the early 1990s.

To comply with the Reviewer's suggestion, we turned to a professional statistician who showed us a different test, which is more appropriate forthis type of research.We carried out another statistical analysis using the recommended test, and its results have been presened in the revised version.

Table 2 shows the different types of anomalies induced by colchicine. We expressed the share of each type of anomalies among all the observed abnormalities in percentage (n = 71). Since we did not find specimens with anomalies in the control group, there is no column for the control in Table 2.

As suggested by the Reviewer, we expressed the information in lines 156-169 in a concise form.

The selection of the dose of colchicine was prompted by the results of previous similar studies of Argas reflexus. The application of the same doses allowed us to compare the effect of this substance on different species of ticks.

Water was the solvent for colchicine; therefore, only water was applied to the control eggs.

Sincerely Yours

Round 2

Reviewer 2 Report

I have now reviewed the paper and it has been revised sufficiently to merit publication.